# Association of the PROGINS PgR polymorphism with susceptibility to female reproductive cancer: A meta-analysis of 30 studies

**Chen Zhou[1,2], Xiangman Zou[2], Xiaosha Wen[2], Zifen Guo[2]***

**1** The Affiliated Nanhua Hospital, Department of Pharmacy, Hengyang Medical School, Unversity of South China, Hengyang, Hunan, 421001, China, **2** Institute of Pharmacy and Pharmacology, Hunan Province Cooperative Innovation Center for Molecular Target New Drugs Study, Hunan Provincial Key Laboratory of Tumor Microenvironment Responsive Drug Research, University of South China, Hengyang, 421001, China

* guozifen76@163.com

**Data Availability Statement:** All relevant data are within the paper and its Supporting Information files.

## Abstract

### Aims

The progesterone response of the nuclear progesterone receptor plays an important role in the female reproductive system. Changes in the function of the progesterone receptor gene may increase the risk of reproductive cancer. The present study performed a meta-analysis to examine whether the progesterone receptor gene *PROGINS* polymorphism was a susceptibility factor for female reproductive cancer.

### Materials and methods

We searched the PubMed, Cochrane Library, Web of Science and EMBASE databases for literature on PROGINS polymorphisms and female reproductive cancer published before September 2020. We evaluated the risk using odds ratios [ORs] and 95% confidence intervals via fixed effects models and random-effects models, which were calculated for all five genetic models. We grouped the analyses by race, cancer, and HWE.

### Results

Thirty studies comprised of 25405 controls and 19253 female reproductive cancer cases were included in this meta-analysis. We observed that the Alu insertion polymorphism and the V660L polymorphism were significantly associated with female reproductive cancer in the allele and dominant genetic models. The allele genetic model and (Alu-insertion polymorphism: OR = 1.22, 95% CI = 1.02–1.45; V660L polymorphism: OR = 1.02, 95% CI = 1.00–1.13) dominant genetic model (Alu-insertion polymorphism: OR = 1.27, 95% CI = 1.03–1.58; V660L polymorphism: OR = 1.10, 95% CI = 1.011.19) demonstrated a significantly increased risk of female reproductive cancer. A subgroup analysis according to ethnicity found that the Alu insertion was associated with female reproductive cancer incidence in white (Allele model: OR = 1.21, 95% CI = 1.00–1.45; Heterozygous model: OR = 3.44,

**Funding:** This study was funded by the Research Fund Project of the Education Bureau of Hunan Province, China (Grant No.19A419) with a grant of CNY 80,000. The funders had no role in study design, data collection and analysis, decision to publish, or preparation of the manuscript.

**Competing interests:** The authors have declared that no competing interests exist.

95% CI = 1.30–9.09) and Asian (Dominant model: OR = 3.12, 95% CI = 1.25–7.79) populations, but the association disappeared for African and mixed racial groups. However, the V660L polymorphism was significantly associated with female reproductive cancer in the African (Allele model: OR = 2.52, 95% CI = 1.14–5.56; Heterozygous model: OR = 2.83, 95% CI = 1.26–6.35) and mixed racial groups (Dominant model: OR = 1.28, 95% CI = 1.01–1.62). Subgroup analysis by cancer showed that the PROGINS polymorphism increased the risk of cancer in the allele model, dominant mode and heterozygous model, but the confidence interval for this result spanned 1 and was not statistically significant. This sensitivity was verified in studies with HWE greater than 0.5.

## Conclusion

Our meta-analysis showed that the progesterone receptor gene Alu insertion and the V660L polymorphism contained in the PROGINS polymorphism were susceptibility factors for female reproductive cancer.

## Introduction

Cancer is a major public health problem worldwide. Cancer is a multifactorial disease, and there is a coordinated relationship between genetic and environmental factors [1,2]. Despite extensive research to prevent cancer, cancer cases continue to increase sharply. Data from the American Cancer Society in 2022 predicts 1.9 million new cancer cases in 2022. More than 609,360 Americans die of cancer annually, which is equivalent to greater than 1,700 people dying of cancer daily [3].

Progesterone is a key regulatory factor in the proliferation and differentiation of female reproductive tract cells. Progesterone inhibits the proliferation of reproductive tract cells by excessive estrogen via the progesterone receptor (PgR) [4–6], and excessive estrogen stimulation increases the risk of female reproductive tract cancer [7]. *PgR* is a member of the nuclear steroid hormone receptor family and is expressed primarily in female reproductive tissues and the central nervous system. It is encoded by a single gene (Gene ID: 5241) located at 11q22–q23 [8], which encodes two isoforms, PgR-B and PgR-A. The two PgR isoforms with different functions come from different transcriptional promoters. PgR-B (114 KDa) is a transcriptional activator and a mediator of cell proliferation, and PgR-A (94 KDa) is a suppressor of transcription. In vitro studies showed that PgR isoforms exhibited different transcriptional regulatory activities. Robert A. et al. [9] found that selective PgR-A knockout induced endometrial epithelial cell proliferation in mice, which suggests that PgR-A is required to control potential adverse reactions of PgR-B. The expression of PgR-A in PgR-B knockout mice was sufficient and necessary to regulate the antiproliferative response of progesterone and estrogen-induced hyperplasia. Prompt changes in the relative expression of these two isoforms or changes in isoform activity or any other genetic mutations may lead to progesterone receptor alienation. The anti-estrogen proliferation effect of progesterone primarily depends on PgR-B, but the excessive expression of PgR-B causes progesterone-dependent proliferation. Progesterone receptor alienation leads to increased susceptibility to female reproductive cancer.

Silencing or mutation of the PgR gene affects the expression of the progesterone receptor. Six variable sites, four polymorphisms, and five common haploids have been detected in the PgR gene. PROGINS contains the Alu insertion in intron 7 of the *PgR* gene, which is

completely linked to the unbalanced linkage (LD) with rs1042838 (V660L in exon 4) and rs1042839 (H770H in exon 5) [10]. The alleles of Alu-insertion alter transcript levels and may contribute to disease risk [11]. The PROGINS polymorphism of the human progesterone receptor diminishes the response to progesterone [12]. The PROGINS allele was significantly associated with decreased serum progesterone levels in patients with polycystic ovary syndrome (PCOS) [13].

The current study considered PROGINS as a risk modifier for gynecological benign and malignant diseases, which indicated that PROGINS may affect *PgR* function. Alu insertion of the PROGINS allele was inversely associated with breast cancer risk and ovarian cancer risk in certain races [14–16]. However, only two research reports that concluded that PROGINS affected the risk of endometrial cancer [17,18]. The V660L polymorphism is caused by G > T, which causes a valine > leucine substitution in the fourth exon of the *PgR* gene. No significant association of the PROGINS polymorphism was found in breast or ovarian cancer studies [19,20]. One study on ovarian cancer [21] also failed to find a link, but another study showed that the T allele (leucine) was associated with an increased risk of breast cancer [22]. However, the results of these studies are inconclusive. Therefore, to clarify the role of the PRPGINS PgR polymorphism in female reproductive cancers, we performed a meta-analysis of all eligible case–control studies to derive the overall cancer risk associated with this polymorphism.

## Materials and methods

The current study conformed to the checklist for meta-analysis of genetic association studies specified for the *PLOS One* approach (S1 Table).

### Literature search and identification

This meta-analysis adhered to the PRISMA guidelines. PubMed, Cochrane Library, Web of Science and EMBASE were used to perform a comprehensive search of published related documents. The following search keywords were used: "polymorphism, genetic" or "breast cancer" or "ovarian cancer" or "endometrial cancer" or "gynecologic neoplasm" and "PROGINS" or "V660 L" or "rs1042838" or "rs1042839" or "H770H" or "Z49816.1" or "Alu-insertion". A search strategy was developed (S2). The last search was updated on September 26, 2020.

### Inclusion and exclusion criteria

The studies were selected using the following criteria.

The following inclusion criteria were used: (a) case-control or cohort study; (b) assessment of PgR polymorphisms for PROGINS and cancer risk; (c) pathology for diagnosis of cancer patients and confirmation that the control was cancer-free; (d) report odds ratios (ORs) and 95% confidence interval (CIs) values or sufficient data to calculate these values; (e) clearly describe genotyping and statistical methods; (f) participants in the control group were in Hardy-Weinberg Balance (HWE); and (g) no language limitations, regardless of sample size.

The following exclusion criteria were used: (a) case reports, comments, comments, and editorial articles; (b) studies of research progress, severity, treatment response, or survival; (c) when overlapping data from the same case series were included in multiple publications, the most recent or most complete study was selected to perform the meta-analysis, and if no information was available, the study was excluded; and (d) literature with specific requirements for the inclusion of cases or. Any differences in the inclusion of the study were resolved via discussion and subsequent consensus.

## Data extraction

Two authors independently extracted the characteristics of the selected study using a standardized protocol, and the third investigator reviewed the results. The following information was extracted from each study: first author, year of publication, study population (country, ethnicity), type of cancer, number of cases and controls, genotype frequencies for cases and controls, and Hardy-Weinberg equilibrium in controls (HWE). We compared key research characteristics, such as location, study time, and authorship, to determine the existence of multiple publications in the same study.

## Quality assessment of the studies

We evaluated the quality based on the NOS quality evaluation to determine the quality of the included literature, and low-quality articles with less than 3 points were excluded. Chen Zhou and Xiangman Zou independently performed the literature search and data extraction. Disputes were discussed and resolved by Xiaosha Wen and Zifen Guo.

## Statistical analysis

STATA software (version 14.0) was used to synthesize the relevant data, and the odds ratio (OR) and 95% confidence interval (CI) were used to evaluate the relationships between PROGINS gene polymorphisms and female reproductive cancer. Five genetic models were used: T2 vs. T1 (allelic), T1T2+T2T2 vs. T1T1 (dominant), T1T2 vs. T1T1 (heterozygous), T2T2 vs. T1T2+T1T1 (recessive), and T2T2 vs. T1T1 (homozygous). Heterogeneity was evaluated using $I^2$ statistics. When the heterogeneity test found significant heterogeneity ($I^2 > 50$ or $P < 0.05$), a random-effect model was used. Otherwise, the fixed model was used. When heterogeneity was present in the study, subgroup analysis was performed according to ethnicity, type of disease and HWE of the included cases to examine the sources of heterogeneity. Sensitivity analysis (excluding one study at a time or changing the model) was used to assess the stability of each efficacy index. Begg's funnel chart and Egger's test were used to evaluate the publication bias of this study. When $P < 0.05$, publication bias was present in this study.

# Results

## Study selection and characteristics

Fig 1 outlines the study selection process in a flowchart following the Preferred Reporting Items for Systematic Reviews and Meta-Analyses (PRISMA) guidelines. A total of 189 articles related to PROGINS polymorphism were retrieved using the retrieval method. Among these articles, 139 articles were excluded after review of the abstracts and unrelated literature, and 11 articles were excluded strictly according to the inclusion and exclusion criteria. Ultimately, 30 articles were included in the meta-analysis (Fig 1) [8,18,19,21–47]. Of the 30 independent studies, 28 studies included genetic frequency analysis of whites, 2 studies included mixed races, 2 studies included Asians and 2 studies included Africans. The types of diseases included breast cancer, ovarian cancer and endometrial cancer. All samples were taken from humans and genotyped using polymerase chain reaction-restriction fragment length polymorphism (PCR–RFLP), DNA sequencing, TaqMan and other genotyping methods (Table 1). The quality of the studies is shown in Table 2.

## Alu-insertion polymorphism and the risk of female reproductive cancer

We counted the Alu-insertion locus and the susceptibility to female reproductive cancer in the five models of allele genetic model (T2 vs. T1), homozygous genetic model (T2T2 vs. T1T2),

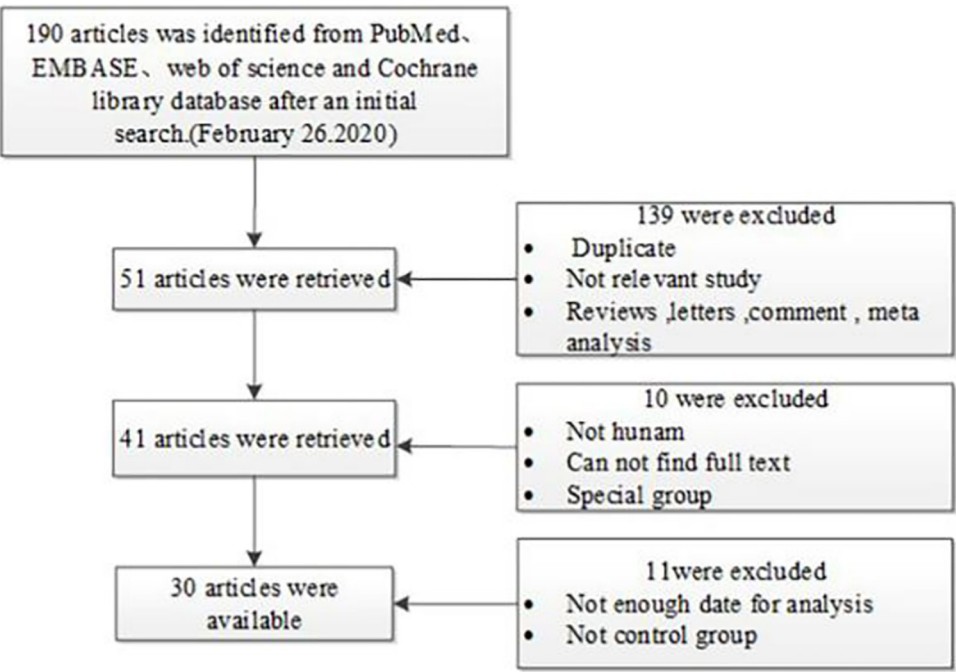

**Fig 1. Flowchart showing the meta-analysis literature screening process.**

heterozygous genetic model (T1T2 vs. T1T1), dominant genetic model (T1T2+T2T2 vs. T1T1) and recessive genetic models (T2T2 vs. T1T2+T1T1) (Table 3). The meta-analysis showed a significant association between Alu-insertion polymorphisms and the risk of female reproductive cancer in the allele genetic model (OR = 1.22 95% CI = 1.02–1.45), the dominant genetic model (OR = 1.27 95% CI = 1.03–1.58), and the heterozygote genetic model (OR = 1.19 95% CI = 1.03–1.38) (Figs 2 and 3). A significant association was found in the allele genetic model of the white group (OR = 1.21 95% CI = 1.00–1.45) (Table 4).

## Val 660 Leu polymorphism and the risk of female reproductive cancer

A total of 16685 cancer patients and 22577 healthy women in 18 studies were used to assess the relationship between the V660 locus and female reproductive cancer risk using the allele genetic model (L vs. V), homozygous genetic model (LL vs. VV), heterozygous genetic model (VL vs. VV), dominant genetic model (VL+LL vs. VV) and recessive genetic models (LL vs. VL+VV). The V660L mutation increased the risk of female reproductive cancer in the allele genetic model (OR = 1.02 95% CI = 1.00–1.13) and dominant genetic model (OR = 1.10 95% CI = 1.01–1.19). The heterozygote genetic model confirmed (OR = 1.09 95% CI = 1.00–1.18) that the V660L mutation increased the risk of female reproductive cancer (Fig 3 and Table 3).

The subgroup analysis found a significant association under the dominant genetic model (OR = 1.21 95% CI = 1.00–1.45) of the breast cancer group (Table 4).

## Publication bias

Begg's and Egger's analyses showed that no publication bias in the Alu insertion or V660L (Table 3).

**Table 1. Characteristics of the studies included in the meta-analysis.**

| Study | Country | Ethnicity | Cancer | Detection method | Sample size (case/control) | Genotype frequency | | | | | | Allele frequency | | | |
|---|---|---|---|---|---|---|---|---|---|---|---|---|---|---|---|
| | | | | | | case | | | controls | | | Case (%) | | Control (%) | |
| | | | | | | $T_1T_1$ | $T_1T_2$ | $T_2T_2$ | $T_1T_1$ | $T_1T_2$ | $T_2T_2$ | $T_1$ | $T_2$ | $T_1$ | $T_2$ |
| Alu insertion | | | | | | | | | | | | | | | |
| Albalawi, I. A | Saudi Arabia | Asian | BC | PCR–RFLP | 100/100 | 81 | 18 | 1 | 93 | 6 | 1 | 10 | 90 | 4 | 96 |
| Donaldson, C J | USA | white | BC | PCR–RFLP | 23/60 | 17 | 5 | 1 | 41 | 16 | 3 | 84.4 | 15.2 | 81.3 | 18.3 |
| Donaldson, C J -2 | USA | African | BC | PCR–RFLP | 61/81 | 56 | 5 | 0 | 73 | 8 | 0 | 95.9 | 4.1 | 95.1 | 4.9 |
| Govindan, S | India | white | BC | PCR–RFLP | 157/108 | 134 | 23 | 0 | 102 | 6 | 0 | 92.7 | 7.3 | 97.2 | 2.8 |
| Junqueira, M. G | Brazil | white | EC | PCR–RFLP | 282/121 | 221 | 61 | 0 | 100 | 18 | 3 | 89.2 | 10.8 | 90.1 | 9.9 |
| Lancaster, J. M -2 | USA | white | BC | PCR–RFLP | 68/101 | 55 | 12 | 1 | 79 | 18 | 4 | 89.7 | 10.3 | 87.1 | 12.9 |
| Lancaster, J. M | USA | white | OC | PCR–RFLP | 309/397 | 219 | 80 | 10 | 285 | 95 | 17 | 83.8 | 16.2 | 83.8 | 16.2 |
| Leite, D.B | Brazil | white | OC | PCR–RFLP | 80/282 | 57 | 12 | 11 | 221 | 61 | 0 | 78.8 | 21.2 | 89.2 | 10.8 |
| Manolitsas, T. P | UK | white | BC | PCR–RFLP | 292/220 | 229 | 61 | 2 | 162 | 54 | 4 | 88.9 | 11.1 | 85.9 | 14.1 |
| Manolitsas, T. P -2 | UK | white | OC | PCR | 231/220 | 173 | 52 | 6 | 162 | 54 | 4 | 86.1 | 13.9 | 85.9 | 14.1 |
| McKenna, N. J | Ireland | white | OC | S-blot | 41/83 | 26 | 15 | 0 | 58 | 21 | 4 | 81.7 | 18.3 | 82.5 | 17.5 |
| McKenna, N. J-2 | Germany | white | OC | S-blot | 26/101 | 17 | 8 | 1 | 88 | 12 | 1 | 80.8 | 19.2 | 93.1 | 6.9 |
| Patricia, G. A | Mexico | white | BC | PCR | 481/209 | 360 | 103 | 18 | 176 | 33 | 0 | 85.6 | 14.4 | 92.1 | 7.9 |
| Runnebaum, I. B | USA | white | OC | PCR | 167/496 | 101 | 60 | 6 | 328 | 153 | 15 | 78.4 | 21.6 | 81.6 | 18.4 |
| Surekha, S | India | white | BC | PCR | 250/249 | 241 | 7 | 2 | 242 | 7 | 0 | 97.8 | 2.2 | 98.6 | 1.4 |
| V660L | | | | | | GG | GT | TT | GG | GT | TT | G | T | G | T |
| Gabriel, C. A | USA | white | BC | TaqMan | 346/357 | 236 | 101 | 9 | 255 | 92 | 10 | 82.8 | 17.2 | 84.3 | 15.7 |
| Gabriel, C. A -2 | USA | African | BC | TaqMan | 86/327 | 75 | 11 | 0 | 309 | 16 | 2 | 93.6 | 6.4 | 96.9 | 3.1 |
| Clendenen, T | Mix | white | BC | TaqMan | 658/1099 | 846 | 288 | 26 | 1516 | 523 | 54 | 85.5 | 14.5 | 84.9 | 15.1 |
| Fabjani, G | Austria | white | BC | DNA | 155/106 | 119 | 32 | 4 | 78 | 28 | 0 | 87.1 | 12.9 | 86.8 | 13.2 |
| Fernandez, L.P | Spain | white | BC | TaqMan | 550/564 | 354 | 153 | 24 | 375 | 154 | 15 | 81.1 | 18.9 | 83.1 | 16.9 |
| Johnatty, S.E | Australia | white | BC | PCR–RFLP | 1444/583 | 1017 | 380 | 47 | 409 | 160 | 14 | 83.6 | 16.4 | 83.9 | 16.1 |
| Lee, E | USA | MIX | EC | TaqMan | 198/1077 | 170 | 25 | 3 | 954 | 114 | 9 | 92.2 | 7.8 | 93.9 | 6.1 |
| Lee, E -2 | USA | white | EC | TaqMan | 379/836 | 259 | 109 | 11 | 615 | 199 | 22 | 86 | 14 | 90.2 | 9.8 |
| Lundin, E | MIX | white | EC | TaqMan | 391/705 | 281 | 96 | 14 | 540 | 147 | 18 | 84.1 | 15.9 | 87 | 13 |
| O'Mara, T. A | Singapore | Asian | EC | TaqMan | 528/1538 | 414 | 151 | 17 | 1147 | 361 | 30 | 84.1 | 15.9 | 86.3 | 13.7 |
| O'Mara, T. A -2 | UK | white | EC | TaqMan | 1086/1591 | 765 | 294 | 27 | 1123 | 434 | 34 | 84 | 16 | 84.2 | 15.8 |
| O'Mara, T. A -3 | Australia | white | EC | TaqMan | 1220/1354 | 867 | 323 | 30 | 933 | 383 | 38 | 84.3 | 15.7 | 83.1 | 16.9 |
| Pearce, C. L | USA | white | OC | DNA | 267/397 | 173 | 82 | 12 | 279 | 111 | 6 | 80.1 | 19.9 | 84.5 | 15.5 |
| Pearce, C. L-2 | USA | white | BC | DNA | 1715/2505 | 1400 | 252 | 15 | 2025 | 363 | 37 | 91.5 | 8.5 | 91 | 9 |
| Pooley, K. A | Englishman | white | BC | TaqMan | 2345/2281 | 1302 | 517 | 42 | 1461 | 513 | 39 | 83.9 | 16.1 | 85.3 | 14.7 |
| Romano, A | Netherlands | white | BC | PCR–RFLP | 167/31 | 123 | 41 | 3 | 22 | 7 | 2 | 85.9 | 14.1 | 82.3 | 17.7 |
| Romano, A | German | white | BC | PCR–RFLP | 545/443 | 399 | 133 | 14 | 347 | 87 | 9 | 85.4 | 14.6 | 88.1 | 11.9 |
| Romano, A -2 | German | white | OC | PCR–RFLP | 67/443 | 42 | 24 | 1 | 347 | 87 | 9 | 80.6 | 19.4 | 88.1 | 11.9 |
| Spurdle, A. B | Australia | white | OC | PCR–RFLP | 551/298 | 395 | 144 | 12 | 203 | 90 | 5 | 84.8 | 15.2 | 83.2 | 16.8 |
| Terry, K. L | USA | white | OC | TaqMan | 895/939 | 648 | 223 | 25 | 612 | 298 | 29 | 84.9 | 15.1 | 81 | 19 |
| De vivo, I | USA | white | BC | TaqMan | 1252/1660 | 869 | 348 | 35 | 1186 | 434 | 40 | 83.3 | 16.7 | 84.5 | 15.5 |
| Tong, D | Austrian | white | OC | DNA | 226/194 | 167 | 50 | 9 | 141 | 52 | 1 | 85 | 15 | 86.1 | 13.9 |
| Quaye, L | UK/USA | white | OC | TaqMan | 1424/2408 | 1005 | 377 | 42 | 1819 | 526 | 63 | 83.8 | 16.2 | 86.5 | 13.5 |
| Ghali, R. M | Tunisia | white | BC | TaqMan | 183/216 | 127 | 50 | 6 | 172 | 37 | 7 | 83.1 | 16.9 | 88.2 | 11.8 |

PCC, population-based case–control study, HCC, hospital-based case–control study, PCR–RFLP PCR-restriction fragment length polymorphism, BC, Breast cancer, EC, Endometrial cancer, OC, Ovarian cancer, DNA, DNA sequencing, S-blot, Southern blot.

**Table 2. Article quality evaluation.**

| Study | Adequate case definition | Definition of controls | Comparability | HWE>0.05 | PCC | PMH (Past Medical History) | Unified detection method | Article quality |
|---|---|---|---|---|---|---|---|---|
| Alu insertion | | | | | | | | |
| Albalawi 2020 [47] | 1 | 1 | 1 | 0 | 0 | 1 | 1 | 5 |
| Donaldson 2002 [28] | 1 | 1 | 1 | 1 | 0 | 0 | 1 | 5 |
| Govindan, S 2007 [34] | 1 | 1 | 1 | 1 | 0 | 0 | 1 | 5 |
| Junqueira, M.G 2007 [18] | 1 | 1 | 1 | 1 | 0 | 1 | 1 | 6 |
| Lancaster 1998 [25] | 1 | 0 | 0 | 0 | 0 | 0 | 1 | 2 |
| Lancaster, J.M 2003 [30] | 1 | 1 | 1 | 0 | 1 | 1 | 1 | 6 |
| Leite, D.B. 2008 [37] | 1 | 1 | 1 | 0 | 0 | 1 | 1 | 5 |
| Manolitsas TP 1997 [24] | 1 | 0 | 1 | 1 | 1 | 0 | 1 | 5 |
| McKenna 1995 [23] | 0 | 0 | 1 | 1 | 0 | 0 | 1 | 3 |
| Patricia Gallegos-Arreola, M 2015 [15] | 1 | 1 | 1 | 1 | 1 | 1 | 1 | 7 |
| Runnebaum, I.B 2001 [26] | 1 | 1 | 1 | 1 | 1 | 1 | 1 | 7 |
| Surekha 2009 [39] | 1 | 1 | 1 | 1 | 1 | 1 | 1 | 7 |
| V660L | | | | | | | | |
| Gabriel, C. A.2013 [44] | 1 | 1 | 1 | 1 | 0 | 1 | 1 | 6 |
| Clendenen, T 2013 [43] | 1 | 1 | 1 | 1 | 1 | 1 | 1 | 7 |
| Fabjani, G 2002 [29] | 1 | 1 | 1 | 1 | 1 | 1 | 1 | 7 |
| Fernandez, L.P 2006 [31] | 1 | 1 | 1 | 1 | 0 | 1 | 1 | 6 |
| Johnatty, S.E 2008 [36] | 1 | 1 | 1 | 1 | 1 | 1 | 1 | 7 |
| Lee, E 2010 [40] | 1 | 1 | 1 | 0 | 1 | 1 | 1 | 6 |
| Lundin, E 2012 [42] | 1 | 1 | 1 | 0 | 1 | 1 | 1 | 6 |
| O'Mara, T.A 2011 [41] | 1 | 1 | 1 | 1 | 1 | 1 | 1 | 7 |
| Pearce, C.L. 2005 [22] | 1 | 1 | 1 | 1 | 1 | 0 | 1 | 6 |
| Pooley, K.A 2006 [32] | 1 | 1 | 1 | 1 | 1 | 0 | 1 | 6 |
| Romano A 2007 [12] | 1 | 1 | 1 | 1 | 1 | 1 | 1 | 7 |
| Romano, A 2006 [33] | 1 | 1 | 1 | 1 | 1 | 0 | 1 | 6 |
| Spurdle 2001 [21] | 1 | 1 | 1 | 1 | 1 | 1 | 1 | 7 |
| Terry, K.L. 2005 [8] | 1 | 1 | 1 | 1 | 1 | 1 | 1 | 7 |
| De vivo 2004 [19] | 1 | 1 | 1 | 1 | 1 | 0 | 1 | 6 |
| Tong, D. 2001 [27] | 1 | 1 | 1 | 1 | 1 | 0 | 1 | 6 |
| Quaye 2009 [38] | 1 | 1 | 1 | 0 | 1 | 1 | 1 | 6 |
| Ghali RM 2020 [46] | 1 | 1 | 1 | 0 | 0 | 1 | 1 | 5 |

low quality, <3; Medium quality,3–4; high quality, ≥5.

## Sensitivity and heterogeneity

A sensitivity analysis was performed to determine whether changes in the inclusion criteria for meta-analysis affected the final results. The author deleted individual studies involved in each meta-analysis to reflect the impact of a single dataset on the merged ORs. Most of the corresponding merged ORs did not change substantially (data not shown). We also changed the effect model to test the impact on the results, and no substantial changes were found on the combined OR, which showed that our results were statistically robust. $I^2$ statistics were used to test the heterogeneity (Table 3), and no heterogeneity was observed in any of the genetic models.

**Table 3. Meta-analysis of the association between the PROGINS polymorphism and female reproductive cancer susceptibility.**

| Polymorphism | Genetic model | Case/Control | Test of association | | Heterogeneity | | | Publication bias | |
|---|---|---|---|---|---|---|---|---|---|
| | | | OR(95%CI) | P | I²(%) | PHet | Model | Egger's test p value | Begg's test p value |
| Alu insertion | T2 vs. T1 | 2568/2828 | 1.22[1.02,1.45] | 0.027 * | 21.3 | 0.218 | F | 0.373 | 0.168 |
| | T2T2+T1T2 vs.T1T1 | 2568/2828 | 1.27[1.03,1.58] | 0.023* | 43 | 0.035 | R | 0.373 | 0.201 |
| | T2T2 vs.T1T2+T1T1 | 2568/2828 | 1.18[0.55,2.55] | 0.670 | 52.1 | 0.015 | R | 0.360 | 0.469 |
| | T2T2 vs.T1T1 | 2046/2322 | 1.23[0.57,2.65] | 0.605 | 52.2 | 0.014 | R | 0.428 | 0.455 |
| | T1T2 vs.T1T1 | 2509/2772 | 1.19[1.03,1.38] | 0.019* | 39 | 0.058 | F | 0.428 | 0.263 |
| V660L | L vs. V | 16685/22577 | 1.07[1.00,1.13] | 0.031* | 15 | 0.253 | F | 0.130 | 0.081 |
| | LL+VL vs. VV | 16685/22577 | 1.10[1.01,1.19] | 0.027* | 60.1 | 0.000 | R | 0.503 | 0.265 |
| | LL vs. VL+VV | 16685/22577 | 1.13[0.99,1.29] | 0.075 | 0 | 0.476 | F | 0.413 | 0.244 |
| | LL vs. VV | 12481/17361 | 1.07[0.93,1.23] | 0.325 | 5 | 0.392 | F | 0.673 | 0.219 |
| | VL vs. VV | 16257/22084 | 1.09[1.00,1.18] | 0.056 | 60.7 | 0.000 | R | 0.385 | 0.204 |

F: Fixed model, R: Random model.

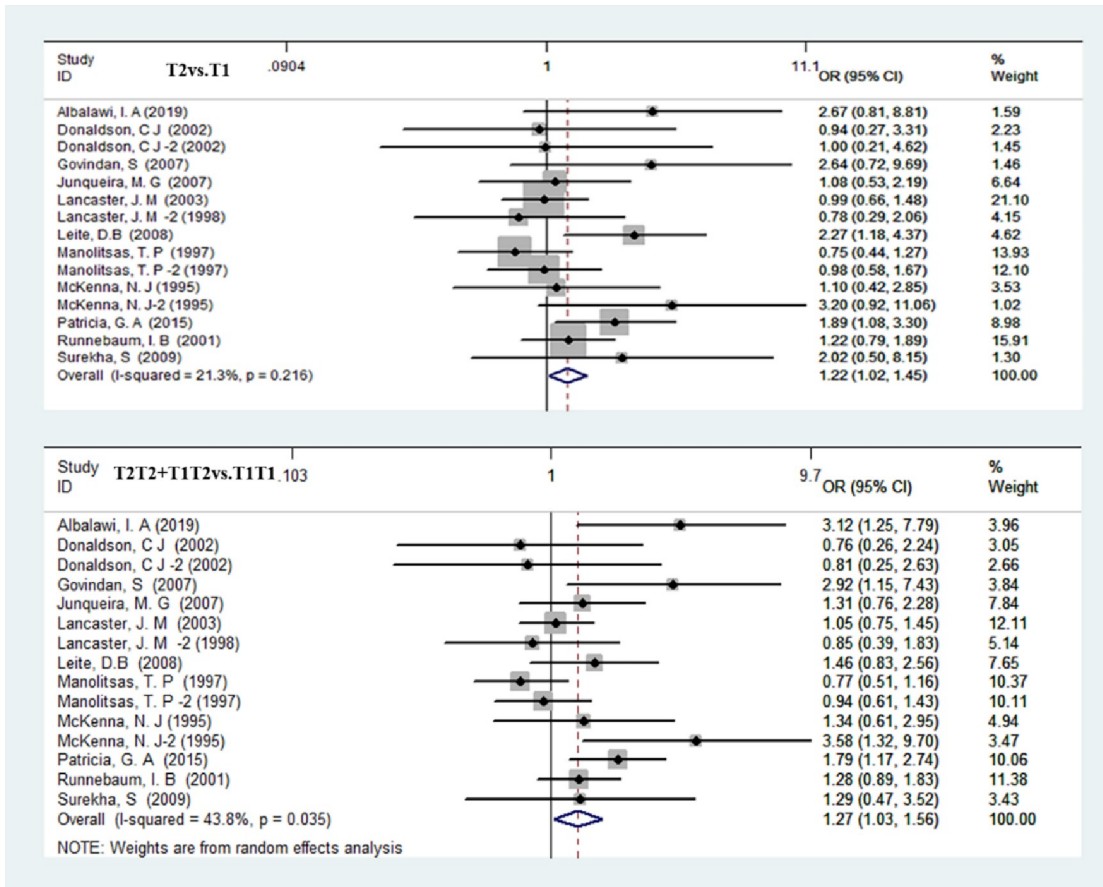

**Fig 2. Forest plot of overall cancer risk associated with Alu-insertion PgR polymorphism (T2 vs. T1 and T2T2+T1T2vs. T1T1).**

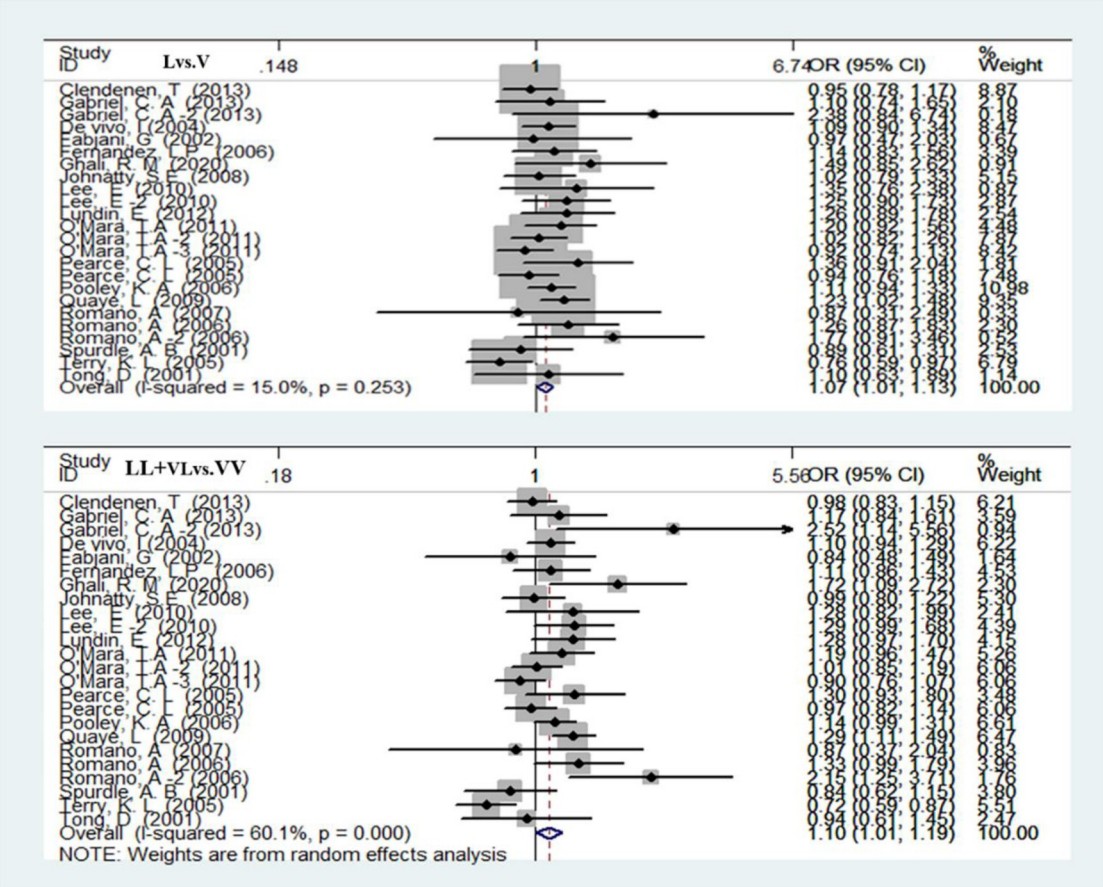

**Fig 3. Forest plot of overall cancer risk associated with V660L PgR polymorphism (L vs. V and LL+VL vs. VV).**

## Discussion

Current evidence indicates that progesterone plays a vital role in regulating female reproduction. The physiological role of progesterone is mediated by the progesterone receptor (PgR), which includes a total of 8 exons and 7 introns [48]. PgR-A and PgR-B are the two subtypes of PgR. The co-expression levels in most normal progesterone-targeted cells are similar. The balance between subtypes regulates the expression of many other genes. Abnormal expression of PgR-A or PgR-B causes a significant change in the ratio between subtypes, which leads to changes in the transmission of progesterone information, and these changes affect physiological functions and trigger a series of serious physiological consequences. The Alu insertion together with V660L and H770H is called PROGINS, which is an important polymorphism of the *PgR* gene. The Alu insertion affects the binding properties of receptors and hormones and induces amino acid changes, which cause female reproductive cancer.

Our meta-analysis included 30 studies with 25405 controls and 19253 female reproductive cancer cases. These studies examined the relationship between *PgR* gene PROGINS polymorphisms (Alu insertion and V660L) and female reproductive tract cancer. Our meta-analysis results demonstrated a significant association between PROGINS and female reproductive cancer, and PROGINS mutations increased the risk of female reproductive cancer. We also performed a sensitivity analysis to test the validity of the results, and the results of the meta-analysis were stable. The association between *PgR* mutations and female reproductive cancer

**Table 4. Pooled odds ratios (ORs) in subgroups.**

| SNP/subgroups | No. of study | Allele model | | | Dominant model | | | Recessive model | | | Homozygous model | | | Heterozygous model | | |
|---|---|---|---|---|---|---|---|---|---|---|---|---|---|---|---|---|
| | | OR | 95%CI | P | OR | 95%CI | P | OR | 95%CI | P | OR | 95%CI | P | OR | 95%CI | P |
| Alu insertion | | | | | | | | | | | | | | | | |
| OC | 5 | 1.20 | 0.95–1.51 | 0.125 | 1.23 | 0.96–1.56 | 0.100 | 1.59 | 0.56–4.49 | 0.382 | 1.67 | 0.60–4.69 | 0.330 | 1.13 | 0.93–1.37 | 0.232 |
| BC | 7 | 1.28 | 0.95–1.72 | 0.101 | 1.30 | 0.86–1.97 | 0.219 | 1.14 | 0.32–4.04 | 0.836 | 1.17 | 0.33–4.29 | 0.818 | 1.23 | 0.97–1.55 | 0.086 |
| EC | 1 | 1.08 | 0.53–2.19 | 0.828 | 1.31 | 0.76–2.28 | 0.170 | 0.06 | 0.00–1.17 | 0.063 | 0.06 | 0.00–1.27 | 0.071 | 1.53 | 0.86–2.73 | 0.146 |
| white | 9 | 1.21 | 1.00–1.45 | 0.046* | 1.23 | 0.99–1.54 | 0.066 | 1.40 | 0.62–3.14 | 0.413 | 1.44 | 0.84–3.28 | 0.377 | 1.14 | 0.98–1.33 | 0.099 |
| Asian | 1 | 2.67 | 0.81–8.81 | 0.108 | 3.12 | 1.25–7.79 | 0.015* | 1.00 | 0.06–16.21 | 1 | 1.15 | 0.07–18.65 | 0.923 | 3.44 | 1.30–9.09 | 0.013* |
| African | 1 | 1.00 | 0.21–4.62 | 0.996 | 0.81 | 0.25–2.63 | 0.731 | —— | —— | —— | —— | —— | —— | 0.81 | 0.25–2.63 | 0.731 |
| Mix | 1 | 1.08 | 0.53–2.19 | 0.828 | 1.31 | 0.75–2.28 | 0.329 | 0.06 | 0.00–1.17 | 0.063 | 0.06 | 0.00–1.27 | 0.071 | 1.53 | 0.86–2.73 | 0.019 |
| HWE > 0.05 | 8 | 1.21 | 0.98–1.50 | 0.076 | 1.27 | 0.98–1.65 | 0.066 | 0.85 | 0.38–1.91 | 0.043 | 1.12 | 0.48–2.61 | 0.797 | 1.22 | 1.02–1.45 | 0.026 |
| HWE < 0.05 | 4 | 1.23 | 0.90–1.68 | 0.188 | 1.30 | 0.85–1.98 | 0.230 | 1.85 | 0.22–15.66 | 0.004 | 1.88 | 0.23–15.50 | 0.556 | 1.13 | 0.86–1.48 | 0.379 |
| V660L | | | | | | | | | | | | | | | | |
| OC | 6 | 1.06 | 0.94–1.20 | 0.334 | 1.09 | 0.82–1.45 | 0.566 | 1.24 | 0.94–1.64 | 0.126 | 1.19 | 0.89–1.59 | 0.230 | 1.05 | 0.78–1.43 | 0.733 |
| BC | 10 | 1.06 | 0.98–1.15 | 0.139 | 1.09 | 1.00–1.19 | 0.052 | 1.07 | 0.88–1.29 | 0.518 | 1.00 | 0.82–1.22 | 0.979 | 1.09 | 0.99–1.20 | 0.064 |
| EC | 3 | 1.07 | 0.96–1.20 | 0.215 | 1.10 | 0.97–1.26 | 0.137 | 1.16 | 0.90–1.50 | 0.256 | 1.11 | 0.85–1.44 | 0.461 | 1.09 | 0.96–1.23 | 0.195 |
| white | 17 | 1.06 | 0.99–1.12 | 0.081 | 1.07 | 0.98–1.17 | 0.134 | 1.10 | 0.95–1.27 | 0.189 | 1.06 | 0.91–1.22 | 0.471 | 1.06 | 0.97–1.16 | 0.209 |
| Asian | 1 | 1.20 | 0.92–1.56 | 0.186 | 1.19 | 0.96–1.47 | 0.109 | 1.51 | 0.83–2.76 | 0.179 | 1.35 | 0.73–2.53 | 0.341 | 1.16 | 0.93–1.45 | 0.190 |
| African | 1 | 2.38 | 0.84–6.74 | 0.103 | 2.52 | 1.14–5.56 | 0.022* | 0.75 | 0.04–15.82 | 0.855 | 0.29 | 0.02–6.55 | 0.434 | 2.83 | 1.26–6.35 | 0.012* |
| Mix | 2 | 1.35 | 0.76–2.38 | 0.305 | 1.28 | 1.01–1.62 | 0.041* | 1.49 | 0.80–2.79 | 0.212 | 1.23 | 0.65–2.42 | 0.499 | 1.25 | 0.97–1.60 | 0.081 |
| HWE > 0.05 | 15 | 1.04 | 0.97–1.11 | 0.229 | 1.03 | 0.93–1.14 | 0.003 | 1.17 | 1.00–1.37 | 0.46 | 1.15 | 0.98–1.35 | 0.084 | 1.03 | 0.95–1.13 | 0.452 |
| HWE < 0.05 | 6 | 1.16 | 1.02–1.31 | 0.020 | 1.34 | 1.10–1.64 | 0.003 | 1.01 | 0.76–1.29 | 0.975 | 0.85 | 0.64–1.14 | 0.277 | 1.29 | 1.06–1.57 | 0.009 |

varies between races. The meta-analysis of the dominant genetic model of the Alu-insertion polymorphism showed that women with T2 mutations had a significantly higher risk of developing female reproductive tract cancer than women with T1T1 genotypes in the general population. There was a significant association between Alu insertion and female reproductive cancer in whites (OR = 1.25, 95% CI = 1.01–1.56), but this association disappeared in Asians and Africans. The difference in correlation may be caused by several factors. First, the frequency of Alu insertion is different due to different ethnic groups, different ethnic groups of genetic backgrounds, different lifestyles, and different environmental factors. Second, there are few reports of the locus in Asians and Africans.

Although some studies showed linkage disequilibrium reactions between Alu insertions and V660L, V660L cannot replace Alu insertions in the analysis of genetic polymorphisms based on these meta-analysis data. The disease correlation between the two polymorphisms was different between ethnicities. Alu insertion was associated with female reproductive cancer incidence in white (Allele model: OR = 1.21, 95% CI = 1.00–1.45; Heterozygous model: OR = 3.44, 95% CI = 1.30–9.09) and Asian populations (Dominant model: OR = 3.12, 95% CI = 1.25–7.79), but the association disappeared for African and mixed racial groups.

Our results showed a significant relationship between V660L and the susceptibility to female reproductive cancer in the allele genetic model, dominant genetic model and heterozygous genetic model.

However, our research has some potential limitations. First, studies that met the inclusion criteria or were unpublished may have been missed. Second, although the control group was primarily selected from healthy people, some people did not mention their physiological status or whether they had benign disease. Finally, 26 studies included whites in the ethnic subgroup analysis, but few studies included Asians and Africans. Therefore, the differences in the

associations between different ethnic subgroups should be carefully interpreted. In conclusion, although there are limitations, the results in this article provide significant evidence that PRO-GINS increases the risk of female reproductive cancer.

## Supporting information

**S1 Checklist.**
(DOCX)

**S1 Table.**
(XLSX)

## Author Contributions

**Conceptualization:** Chen Zhou.

**Data curation:** Chen Zhou, Xiangman Zou, Xiaosha Wen.

**Funding acquisition:** Zifen Guo.

**Methodology:** Chen Zhou.

**Writing – original draft:** Chen Zhou.

**Writing – review & editing:** Chen Zhou, Zifen Guo.

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
