## [Decision Letter · Decision Letter 0]

10 Feb 2022

PONE-D-21-14829Association of PROGINS PgR Polymorphism with susceptibility to female reproductive cancer: A meta-analysis based on 30 studiesPLOS ONE

Dear Dr. Guo,

Thank you for submitting your manuscript to PLOS ONE. After careful consideration, two experts considered the manuscript interesting and well described, but it does not fully meet PLOS ONE’s publication criteria as it currently stands. Therefore, we invite you to submit a revised version of the manuscript that addresses the points raised during the review process.

We look forward to receiving your revised manuscript.

Kind regards,

Elda Tagliabue

Academic Editor

PLOS ONE

Journal Requirements:

2. Please include your tables as part of your main manuscript and remove the individual files. Please note that supplementary tables (should remain/ be uploaded) as separate "supporting information" files

3. Please note that according to our submission guidelines (http://journals.plos.org/plosone/s/submission-guidelines), outmoded terms and potentially stigmatizing labels should be changed to more current, acceptable terminology. For example: “Caucasian” should be changed to “white” or “of [Western] European descent” (as appropriate).

4. We suggest you thoroughly copyedit your manuscript for language usage, spelling, and grammar. If you do not know anyone who can help you do this, you may wish to consider employing a professional scientific editing service. 

Whilst you may use any professional scientific editing service of your choice, PLOS has partnered with both American Journal Experts (AJE) and Editage to provide discounted services to PLOS authors. Both organizations have experience helping authors meet PLOS guidelines and can provide language editing, translation, manuscript formatting, and figure formatting to ensure your manuscript meets our submission guidelines. To take advantage of our partnership with AJE, visit the AJE website (http://aje.com/go/plos) for a 15% discount off AJE services. To take advantage of our partnership with Editage, visit the Editage website (www.editage.com) and enter referral code PLOSEDIT for a 15% discount off Editage services.  If the PLOS editorial team finds any language issues in text that either AJE or Editage has edited, the service provider will re-edit the text for free.

A clean copy of the edited manuscript (uploaded as the new *manuscript* file)”"

This study is partially funded by the Natural Science Foundation of Hunan Province (No.2018JJ2350) and the key project of Education Department of Hunan Province (No.19A419).

6. Thank you for stating the following in the Acknowledgments/ Funding Section of your manuscript: 

This study is partially funded by the Natural Science Foundation of Hunan Province (No.2018JJ2350) and the key project of Education Department of Hunan Province (No.19A419).

NO - Include this sentence at the end of your statement: The funders had no role in study design, data collection and analysis, decision to publish, or preparation of the manuscript.

7. Please amend your list of authors on the manuscript to ensure that each author is linked to an affiliation. Authors’ affiliations should reflect the institution where the work was done (if authors moved subsequently, you can also list the new affiliation stating “current affiliation:….” as necessary).

8. Please amend either the abstract on the online submission form (via Edit Submission) or the abstract in the manuscript so that they are identical.

9. Your ethics statement should only appear in the Methods section of your manuscript. If your ethics statement is written in any section besides the Methods, please move it to the Methods section and delete it from any other section. Please ensure that your ethics statement is included in your manuscript, as the ethics statement entered into the online submission form will not be published alongside your manuscript. 

10. Please include captions for your Supporting Information files at the end of your manuscript, and update any in-text citations to match accordingly. Please see our Supporting Information guidelines for more information: http://journals.plos.org/plosone/s/supporting-information. 

11. We note that this manuscript is a systematic review or meta-analysis; our author guidelines therefore require that you use PRISMA guidance to help improve reporting quality of this type of study. Please upload copies of the completed PRISMA checklist as Supporting Information with a file name “PRISMA checklist”.

Reviewers' comments:

Reviewer's Responses to Questions

**Comments to the Author**

1. Is the manuscript technically sound, and do the data support the conclusions?

Reviewer #1: Yes

Reviewer #2: Yes

2. Has the statistical analysis been performed appropriately and rigorously? 

Reviewer #1: Yes

Reviewer #2: Yes

3. Have the authors made all data underlying the findings in their manuscript fully available?

Reviewer #1: Yes

Reviewer #2: Yes

4. Is the manuscript presented in an intelligible fashion and written in standard English?

Reviewer #1: Yes

Reviewer #2: No

5. Review Comments to the Author

Reviewer #1: This study is a meta-analysis of the association between PgR gene polymorphisms and female reproductive cancer. This is a very interesting study, but I would like to point out the following points.

Major comments

１． Most of the papers that authors adopted for this meta-analysis are on cancers that occur in specific organs (breast, endometrial, and ovarian cancer) and gene polymorphisms in PgR. Does the role of PgR play a similar role in each of these organs? Is it justified to perform a meta-analysis together with female reproductive cancer without considering organ specificity?　 For example, past epidemiological studies have shown that hormone replacement therapy, including progesterone, has different effects on breast and endometrial cancer risks.

２． From the viewpoint of Comment 1, the interpretation of the results of this study, which examined the relationship between female reproductive cancer and gene polymorphisms in PgR, is difficult to interpret and seems to have low clinical significance.　 I think it is more clinically significant to focus on organ-specific meta-analysis and discuss whether there are any differences.

Minor moment

３． Abstract

In the Material and Method section, it is said that the analysis was performed for the race, cancer, and HWE groups, but the in section of Result, the description of the analysis result is only for race.

4. Figure 1

Check if the numbers in the study consort are correct.　 The total number of excluded studies and the number of non-excluded studies do not match.

Reviewer #2: Detailed comments for the author's consideration

Review of the paper by �Chen Zhou, Xiangman Zou, Xiaosha Wen， Zifen Guo entitled: “Association of PROGINS PgR Polymorphism with susceptibility to female reproductive cancer: A metaanalysis based on 30 studies” The work presented is interesting in terms of demonstrating the association of PROGINS PgR Polymorphism with susceptibility to female reproductive cancer. The manuscript is well written and results are well described with proper statistics. The language is appropriate, technical and easy to read. However, I have some minor recommendations

Comment 1: The highlights of the manuscript should be written properly with more conclusive meaning.

Comment 2: The first sentence of the abstract is not properly written. It should be rewritten with proper meaning.

Comment 3: The first para of introduction section has 2020 data. Update the para with 2021/22 epidemological and incidence data.

Comment 4: In second para of introduction section there are some sentences related to estrogen which creates confusion. Remove the estrogen related sentences or If not rewrite the sentence with proper meaning. Additionally, in second para “vitro” should be replaced with “invitro”

Comment 5: Add some lines about PROGINS and its functions in para 3 of introduction section.

Comment 6: There are various spelling and grammatical mistakes such as “conformed” in materials and methods section. Kindly go through the whole manuscript to edit grammatical mistakes.

Comment 7: Manuscript contains old and outdated references, replace them with latest and updated references wherever possible.

6. PLOS authors have the option to publish the peer review history of their article (what does this mean?). If published, this will include your full peer review and any attached files.

Reviewer #1: No

Reviewer #2: No

---

## [Author Response · Author response to Decision Letter 0]

11 May 2022

Dear the editor and reviewers: 

Thank you very much for giving us an opportunity to revise our manuscript (PONE-D-21-14829) titled Association of PROGINS PgR Polymorphism with susceptibility to female reproductive cancer: A meta-analysis based on 30 studies. We appreciate the positive and constructive comments and suggestions from the editor and reviewers, which significantly improved the quality of our manuscript. We have carefully revised the manuscript accordingly with all changes shown in red in the paper. Please see the detailed responses below.

Reviewers 1' Comments to Author

This study is a meta-analysis of the association between PgR gene polymorphisms and female reproductive cancer. This is a very interesting study, but I would like to point out the following points.

Comment 1: Most of the papers that authors adopted for this meta-analysis are on cancers that occur in specific organs (breast, endometrial, and ovarian cancer) and gene polymorphisms in PgR. Does the role of PgR play a similar role in each of these organs? Is it justified to perform a meta-analysis together with female reproductive cancer without considering organ specificity? For example, past epidemiological studies have shown that hormone replacement therapy, including progesterone, has different effects on breast and endometrial cancer risks.

RESPONSE: Thank you very much for your constructive suggestions. Progesterone and progesterone receptors (PgR) are essential for the development and cyclical regulation of hormone-responsive tissues including the breast and reproductive tract. We think it is reasonable to conduct a Meta-analysis on female reproductive cancers. Human is an organic whole, and the regulation of genes by genetic polymorphisms is holistic. As a steroid hormone, the effects of progesterone exerts on the female reproductive system are complicated.

The main subtypes of PgR are PgR-A and PgR-B mediating the main progesterone responses, but the roles of PgR-A and Pgr-B in different organs are not the same. PgR-A is essential for ovarian normal function in ovarian cancer, while its expression is decreased or lost in ovarian cancer. PgR-B has the functions of anti-proliferation (such as anti-aging and anti-apoptosis), playing a dominant role in ovarian cancer. In breast cancer, PgR has been shown to rapidly activate the Src/p21Ras/Erk, PI3K/Akt, and JAK/STAT pathways that contribute to the proliferative effect of progesterone in breast cancer cells, but unliganded PgR has been shown to suppress growth and inflammatory responses in breast cancer cells. The PgR status of endometrial tumors has been controversial. One of the previous studies showed the predominance of PgR-B in advanced endometrial tumors, but another study noted the loss of both subtypes in advanced endometrial cancers; additionally, a third study showed that only PgR-A was expressed in poorly differentiated endometrial cancer cell lines. At present, the imbalance between the expression of PgR-A and PgR-B is one of the pathogenesis of endometrial cancer.

Comment 2：From the viewpoint of Comment 1, the interpretation of the results of this study, which examined the relationship between female reproductive cancer and gene polymorphisms in PgR, is difficult to interpret and seems to have low clinical significance. I think it is more clinically significant to focus on organ-specific meta-analysis and discuss whether there are any differences.

RESPONSE: Thank you very much for the good advice. As to the data presented in this paper, the risk propensity for female reproductive cancers is the same as that for ovarian, breast, and endometrial cancers. The data in table 4 suggest that PROGINS gene polymorphisms may be associated with a slightly increased risk of ovarian, breast, and endometrial cancers.

Comment 3： Abstract，In the Material and Method section, it is said that the analysis was performed for the race, cancer, and HWE groups, but the in section of Result, the description of the analysis result is only for race.

RESPONSE: Thank you very much for the good advice. We added the analysis results of cancer and HWE groups to the Abstract.

Abstract： Subgroup analysis by cancer, PROGINS polymorphism increase the risk of cancer in the Allele model, Dominant mode and Heterozygous model, but the confidence interval for this result spanned 1 and was not statistically significant. At the same time, this sensitivity was also verified in studies with HWE greater than 0.5.

Comment 4： Figure 1，Check if the numbers in the study consort are correct.　 The total number of excluded studies and the number of non-excluded studies do not match. 

RESPONSE: Thank you very much for the good advice. We verified the data in Figure 1 and corrected this error.

Fig 1. Flowchart showing the meta-analysis literature screening process

Reviewers 2' Comments to Author:

The work presented is interesting in terms of demonstrating the association of PROGINS PgR Polymorphism with susceptibility to female reproductive cancer. The manuscript is well written and results are well described with proper statistics. The language is appropriate, technical and easy to read. However, I have some minor recommendations.

Comment 1: The highlights of the manuscript should be written properly with more conclusive meaning.

RESPONSE: Thank you very much for the good advice. We have refined the highlights.

Highlights

Alu insertion may be a susceptible factor for reproductive cancer in white and Asian populations, and the V660L polymorphism may be a susceptible factor for female reproductive cancer in African populations.

Comment 2: The first sentence of the abstract is not properly written. It should be rewritten with proper meaning.

RESPONSE: Thank you very much for the good advice. We rewrote this sentence.

Aims: The progesterone response of the nuclear progesterone receptor plays an important role in the female reproductive system.

Comment 3: The first para of introduction section has 2020 data. Update the para with 2021/22 epidemological and incidence data. 

RESPONSE: Thank you very much for the good advice. We have updated the latest epidemiological data for 2022.

Despite extensive research to prevent cancer, cancer cases continue to increase sharply. Data from the American Cancer Society in 2022 predicts 1.9 million new cancer cases in 2022. More than 609,360 Americans die of cancer annually, which is equivalent to greater than 1,700 people dying of cancer daily [3].

Comment 4: In second para of introduction section there are some sentences related to estrogen which creates confusion. Remove the estrogen related sentences or If not rewrite the sentence with proper meaning. Additionally, in second para “vitro” should be replaced with “invitro”.

RESPONSE: Thank you very much for the good advice. We rewrote the sentence to clarify it and replaced the word.

Progesterone inhibits the proliferation of reproductive tract cells by excessive estrogen via the progesterone receptor (PgR) [4-6], and excessive estrogen stimulation increases the risk of female reproductive tract cancer [7]. 

Comment 5: Add some lines about PROGINS and its functions in para 3 of introduction section.

RESPONSE: Thank you very much for the good advice. We added the functions of PROGINS to the Introduction.

The PROGINS polymorphism of the human progesterone receptor diminishes the response to progesterone [12]. The PROGINS allele was significantly associated with decreased serum progesterone levels in patients with polycystic ovary syndrome (PCOS) [13].

Comment 6: There are various spelling and grammatical mistakes such as “conformed” in materials and methods section. Kindly go through the whole manuscript to edit grammatical mistakes.

RESPONSE: Thank you very much for the good advice. We carefully revised the sentences in the article, and we have updated and revised our manuscript through the language editing service from AJE (Verification Code: 0283-9E24-1846-6C24-A673). Certificate of language editing as follows:

Comment 7: Manuscript contains old and outdated references, replace them with latest and updated references wherever possible. 

RESPONSE: Thank you very much for the good advice. We updated the current references, but the references the data extracted from could not be replaced.

---

## [Editor Report · Decision Letter 1]

28 Jun 2022

Association of the PROGINS PgR polymorphism with susceptibility tofemale reproductive cancer: A meta-analysis of 30 studies

PONE-D-21-14829R1

Dear Dr. Guo,

After a careful assessment of the revised version of the manuscript by the Editor, we’re pleased to inform you that your manuscript has been judged scientifically suitable for publication and will be formally accepted for publication once it meets all outstanding technical requirements.

Kind regards,

Elda Tagliabue

Academic Editor

PLOS ONE
---

## [Editor Report · Acceptance letter]

7 Jul 2022

PONE-D-21-14829R1 

Association of the PROGINS PgR polymorphism with susceptibility to female reproductive cancer: A meta-analysis of 30 studies 

Dear Dr. Guo:

I'm pleased to inform you that your manuscript has been deemed suitable for publication in PLOS ONE. Congratulations! Your manuscript is now with our production department. 

Kind regards, 

on behalf of

Dr. Elda Tagliabue 

Academic Editor

PLOS ONE